# Cross-School Collaboration to Develop and Implement Self-Construction Greening Systems for Schools

**DOI:** 10.3390/plants12020327

**Published:** 2023-01-10

**Authors:** Florian Teichmann, Ines Kirchengast, Azra Korjenic

**Affiliations:** Research Unit of Ecological Building Technologies, Institute of Material Technology, Building Physics and Building Ecology, Faculty of Civil and Environmental Engineering, TU Wien—Vienna University of Technology, 1040 Vienna, Austria

**Keywords:** urban green areas, school greening, low-cost greenery systems, construction manuals, interdisciplinary collaboration, student participation

## Abstract

The positive effects of green infrastructure in the urban environment are nowadays widely known and proven by research. Yet, greening, which serves to improve the indoor climate and people’s well-being, is integrated very limited in public facilities such as schools. Reasons for this are seen in a lack of knowledge and financing opportunities. A focus, among others, of the MehrGrüneSchulen research project is the interdisciplinary development of cost-effective greening solutions for schools. The designs were developed in close collaboration with students of a technical college (HTL) and a horticultural school. This study describes the development process and presents the results of the first implementations of greening systems at the HTL-building complex and at nine other schools in Austria.

## 1. Introduction

As a measure for climate change adaptation and the improvement of urban ecology, it has already been demonstrated at various levels of national and international projects that green infrastructure in the built environment has positive effects on the microclimate. The previous school greening projects GRÜNEzukunftSCHULEN [1] and GrünPlusSchule [2] confirm the positive effects in the school sector. Due to the usually high cost of greening systems available on the market, greening in and on buildings, regardless of their use, has only been implemented on a small scale to date. However, public interest in integrating more “green” into the urban environment is increasing.

According to a survey by UNICEF [3], children and young people spend 38.5 h a week at school and doing homework. Hence, good indoor air quality is significant for the well-being and conductive to concentration. If the indoor air quality in a room is poor, well-being can be affected and the performance of the people in it can decrease. Especially in schools as places where children learn and develop, it is of great importance to provide a healthy learning environment and a high indoor air quality. Extensive greening at schools can make an essential contribution to this.

The research project MehrGrüneSchulen, funded by the Climate and Energy Fund and carried out as part of the program “Smart Cities Demo-Living Urban Innovation 2019”, aims to enable more greening solutions at schools in Austria. Simple greening systems that can be implemented with low financial resources are to be developed and made available free of charge in the form of do-it-yourself (DIY) construction manuals. Schools are thus supported in their role as experimental spaces in the city. Throughout the development, an increased independent, practical implementation of green infrastructure at schools will be promoted, based on scientific findings from preliminary projects.

The goal of this study is to reveal the process of developing greening systems within an interdisciplinary teaching format and to present the results of student work and construction and greening workshops. Finally, previous research on school greening is cited and related to the results of this study in order to provide guidance for teachers and advance the integration of greening into regular school operations.

## 2. Results

An important output of the MehrGrüneSchulen research project includes four construction manuals for indoor greening systems and five construction manuals for outdoor greening systems. The systems can be built as described or adapted and further developed as desired. In ten schools, distributed throughout Austria, innovative greening systems were created, which contribute to an improvement of the learning environment and thus offer a sustainable added value for the schools and consequently for society.

At each stage, from conception to implementation, students from different schools and disciplines were involved in order to gain knowledge around the subject, improve their motor skills and increase collaboration. In the following sections, the first designs and prototypical implementations of the developed greening systems are presented and impressions from the construction process are offered. Furthermore, the built results of the implementation workshops in the federal states are presented.

### 2.1. Greening Systems for School Interiors and Facades

The illustrations of the final designs of the facade and interior greening systems, made by students of the Camillo Sitte Building Technology Center, Vienna (subsequently referred to as “CSBT”), are shown in Figure 1. The ideas were developed in collaboration with students of the Higher Federal Teaching and Research Institute for Horticulture and Austrian Federal Gardens at Schönbrunn, Vienna (subsequently referred to as “HBLFA”), who subsequently provided suggestions for the choice of plants (shown only schematically in Figure 1). The design posters of the respective groups of students can be viewed on the project homepage.

The specifics of the developed facade and interior greening systems can be described as follows:Facade greening system “The Green Plug”:This specially shaped molded brick, which can be produced by pressing, can be used to form continuous troughs for a greened clinker facade with tongue and groove joints. The system, like most clinker facades, follows the principle of a ventilated facade, which must be fixed at regular intervals by dowelling to the supporting wall structure. Irrigation can be provided by simple drip hoses.Planting—for facade application: *Hosta tardiana, Fuchsia magellanica, Bergenia cordifolia, Heuchera micrantha, Arum italicum, Brunnera macrophylla; for indoor application: Calathea makoyana, Begonia maculata, Peperomia, Chlorophytum comosum, Monstera adansonii, Syngonium podophyllum*.Clinker brick facade trough:This facade greening system consists of 80 cm high and 100 cm wide brick troughs that can be placed on top of each other in a tongue and groove system. Due to the large troughs, the plants have plenty of root space, which also serves to store water and therefore requires less watering. Here, too, fastening to the outer wall construction is done by facade anchors. Mounting on interior walls is also possible in principle, although in this case it may be advantageous to reduce the dimensions of the troughs.Planting—for facade application: *Festuca glauca, Hylotelephium telephium, Sempervivum, Yucca gloriosa, Cotoneaster dammeri, Campanula poscharskyana, Thymus praecox, Phlox subulate, Hypericum, Rosmarinus officinalis*.Modular wall greening system:This is a modular greening system with a grid-shaped wooden substructure, which is intended for mounting on a load-bearing interior wall. The plant troughs, made of sheet metal, can be hung in any position on the wooden grid structure and vary in their dimensions. This allows for individual adaptability in terms of size and plant density of the overall system.Planting—for indoor application: *Senecio rowleyanus, Callisa repens, Ficus pumila, Ceropegia linearis, Calathea, Peperomia glabella, Chlorophytum comosum, Adiantum capillus-veneris*.Greened seating furniture “Green Domino”:The idea of this greening system is to create a large-scale seating landscape, e.g., for auditoriums or break rooms, by arranging several elements rotated by 90° in a domino-like manner. This invites students to linger in a green oasis and encourages them to change the arrangements of the individual furniture according to their needs. Wheels attached to the bottom allow easy movement of the wooden greening system, which is made without metal connectors.Planting—for indoor application: *Senecio rowleyanus, Syngonium podophyllum, Zantedeschia aethiopica, Epipremnum aureum, Peperomia obtusifolia, Philodendron erubescens, Maranta leuconeura, Aglaonema commutatum, Ficus pumila, Ctenanthe burle-marxii*.Plantable mobile partition:The original design of this system utilizes a used heavy-duty clothes bar, which is converted into a greening system by simple adaptation. For the design at the CSBT’s construction yard, the variant with wooden greening troughs on both sides was chosen. These were lined with pond foil and have an integrated water storage layer, enabling a design without water drainage.Planting—for indoor application: *Bergenia, Spathiphyllum calathea, Adiantum raddianum, Aglaonema commutatum, Anthurium andreanum*.Greened hanging system “Green Cloud”:This flat greening system is suspended from the ceiling if the room is high enough, creating a “green cloud” above the heads of students and teachers. The square wooden grid construction has a metal lattice in the center. The four elongated aluminum troughs are placed in the openings of the wooden structure, allowing the plants to grow over the metal lattice and hang down from the outer sides.Planting—for indoor application: *Chlorophytum, Asparagus densiflorus, Philodendron, Monstera, Tradiscantie, Acalypha hispida, Nematanthus gregarious, Nephrolepis exaltata, Hoya, Epipremnum aureum*.

After completion of the designs, preliminary construction guidelines were developed. This created the basis for implementing the first prototypes at the CSBT construction yard (see Figure 2). Through the practical, partly empirical implementations, further findings and constructive optimizations were incorporated into the completion of the respective building manuals, which were created for as broad a target group as possible. In advance, it must be decided whether the required materials will be prefabricated in their final dimensions by a hardware store, a carpentry shop or similar, or whether the students will carry out the work themselves under supervision. This depends primarily on the infrastructure of the school itself, which, in the case of the CSBT, allowed all the work steps to be carried out in the building yard. The students’ achievements were presented at a public event on-site (see Figure 3). The greening systems have since been in use in the school building.

### 2.2. Greening Systems for School Open Space

From the beginning, the focus was on the climate effectiveness of the systems, which is why the largest possible plant volume was to be provided. In addition, materials should be used that are as cost-efficient, ecological, and easy to process as possible, which is reflected in the dominant use of wood as the primary load-bearing structure. Again, the design posters from the respective student groups are publicly available on the project homepage.

The following six greening systems for the schools’ open spaces were developed (final designs are shown in Figure 4):“GreenClassroom” greened pergola:This spacious pergola allows outdoor classes to be held (the “open space classroom” principle) and can be used for individual purposes during breaks. Due to its modularity, it is possible to adapt the dimensions to the intended class size. Stretching from one wooden pillar to the next, planting troughs with climbing support for various plants are provided to green the pergola as much as possible. The addition of opaque wall and ceiling panels with respect to year-round usability of the pergola has been considered.Planting—for outdoor application: *Clematis alpina, Wisteria floribunda, Actinidia arguta, Vitis vinifera, Rosa lucieae*.Plantable hanging chair “Green Trio”:The name originated from the idea of always arranging three such hanging chairs around a central area, e.g., an existing tree, thus increasing the tilt resistance of the individual hanging chairs in addition to their special appearance. This greening system, like the other systems, is primarily made of wood, but it also has load-bearing steel cables with which the hanging chair is suspended from the supporting structure. The greenery is provided by climbing plants, which are inserted either into the existing soil or into wooden troughs that can be positioned behind the hanging chair.Planting—for outdoor application: *Clerodendrum thomsoniae, Lonicera caprifolium*.Raised bed “The Vessel”:The special features of this raised bed are the integrated seating area and the storage space underneath for all sorts of garden utensils. The construction is made entirely of wood. The walls of the raised bed facing the soil body should be protected from moisture by a waterproofing membrane. The raised bed should be permeable to the soil below to avoid water accumulation and allow soil organisms to bond with the new soil body. A small-meshed grid avoids the entry of voles.Planting—for outdoor application: all kinds of fruits, vegetables and herbs, such as berries, salads, zucchini, squash, onion, rosemary, hyssop, etc.Greened pergola “T-Bench”:This pergola was named for its shape, which resembles a T. In the center of the pergola is a plant trough for the climbing plants in order to cover the roof, providing the desired shade. Benches are placed on either side of the planting trough, hiding the trough and allowing the climbing plants to grow out of the gap between the benches. An alternative design for the T-Bench is to omit the bench on one side and instead provide a place to park bicycles.Planting—for outdoor application: *Vitis silvestris, Hedera helix, Clematis, Phaseolus coccineus, Thunbergia alata, Ipomoea, Tropaeolum*.Pergola with play equipment “Place Evergreen”:Inspired by the shape of a honeycomb, this pergola has six sides of equal length and can thus be extended by another honeycomb element on each side. The design provides for two different modules that can be combined with each other as requested: on the one hand, a playground module with various playground equipment and a climbing net on the roof and, on the other hand, a relaxation module with Holly- wood swings and a green roof.Planting—for outdoor application: *Clematis alpina, Wisteria floribunda, Actinidia arguta, Vitis vinifera, Rosa lucieae*.Greened fountain:This greening system is designed to create a feel-good oasis for hot summer days: In the center is a water area with water plants and a solar fountain. The seating areas next to it are bordered on the back by a green wall of climbing plants on a climbing scaffold. The watering of the climbing plants is solar-powered as well.Planting—for outdoor application: *Euonymus fortunei, Clematis alpina, Lonicera caprifolium, Pygmaea Chrysantha, Hydrocharis morsus-ranae, Hottonia palustris, Stratiotes aloides, Hippuris vulgaris, Veronica beccabunga*.

Of the six open space greening systems, two, the “Green Trio” and “T-Bench” systems, were implemented at the CSBT construction yard (see Figure 5). The “Green Classroom,” “T-Bench,” and “The Vessel” were, among others, implemented at the greening workshops in the federal states (see Section 2.3). The systems “Place Evergreen” and the greened fountain have not yet been implemented due to their complexity.

As with the indoor greening systems, the construction of the outdoor systems requires the availability of appropriate tools and machines as well as the know-how to operate them. In the final elaboration of the construction manuals, the applicability for a broad target group was taken into account. Due to the size and complexity of some of the systems, however, it is advisable to enlist professional support, for example in the handling of wood.

Rebuilding of the proposed greening systems is at the builders’ and users’ own risk, as the building instructions only represent the rough design intentions of the editors and do not take any safety-relevant measures into account. The teachers and supervisors responsible for the construction must be aware of this and must take the necessary safety precautions when working with students. When selecting the installation site, the static, fire protection, building physics and other conditions must be considered and, if necessary, the approval of the building owner must be obtained. For long-term functioning of the system, care should also be taken to ensure suitable lighting, watering and fertilization for the plants used and to replace the plant substrate at regular intervals (approximately every 3 years). It is recommended to hire a specialized company for maintenance and extensive care work. Dead plant parts are to be removed continuously and replaced by new ones if necessary. This work can be done by students or by elected responsible persons.

### 2.3. Results of the Greening Workshops in the Federal States

After completion of the greening systems including the DIY-instruction manuals by CSBT students together with the project team, the individual workshops took place in the selected schools of the federal states. Different greening systems for indoor and outdoor areas were implemented. The results of these greening workshops are shown in Figure 6.

The following greening systems were implemented during the workshops:Korneuburg (Lower Austria):The implemented greening system corresponds to a double version of the pergola “T-Bench”, which was erected on concrete slabs. The climbing plants were inserted directly into the ground. The sides of the benches were covered for visual reasons. For the care of the plants, the backrest on one side of the bench can be removed with only a few screws.Wörgl (Tyrol):The implementation is based on the greened pergola “Green Classroom”, whereby the dimensions were adapted to minimize the wood waste. In one corner, a free space was provided for the integration of a stove, which will be retrofitted by the school.St. Johann (Salzburg):Since the school in St. Johann already had a large-scale climbing trellis on several exterior wall surfaces, but the old climbing plants had already died, the principal decided to use the existing trellis and add new climbing plants. Therefore, no further greening system was implemented here.Maria Gail (Carinthia):For the elementary school in Maria Gail near Villach, the principle of the mobile green partition was adapted and simplified for safety reasons. Instead of the clothes rail the construction was made of wood. Since the construction of such a greening system is not feasible with elementary school children, the wooden construction was prefabricated in Vienna, so that only the fleece, the substrate and the plants had to be inserted at the workshop.Graz (Styria):In the high school in Graz, the same mobile green walls were erected as in Maria Gail. In contrast to the elementary school, however, the wooden construction was built together with students from the upper school, which promoted the interaction of different age groups and school levels.Kirchdorf (Upper Austria):Like Wörgl, a slightly modified version of the “Green Classroom” pergola was built in Kirchdorf. Here, the plant troughs as well as the bench are arranged on one side only, so that people sit with their back to the sports field, with a certain separation provided by the greenery.Neusiedl (Burgenland):At the Pannoneum Neusiedl School of Economics and Tourism, a raised bed for the independent cultivation of fruit and vegetables was built by the students. In contrast to the design of the raised bed “The Vessel”, a bench with storage space underneath was omitted.Lauterach (Vorarlberg):Due to the long distance to Vienna, the process of planning and implementing a school greening at the secondary school in Lauterach was supervised exclusively online. The result of the planning was a building-high greening of a windowless wall with scaffolding climbers. In addition, smaller greening measures were also implemented in the building.

The results of the greening workshops show that the construction of simple systems for the greening of schools can be well implemented within the framework of handicraft lessons. The perquisite is good preparation and planning to achieve the desired greening goal. As can be seen in Figure 6, the systems are largely made of ecological materials and their design relates to the needs and requirements of the schools. The greening measures also contribute to an aesthetic enhancement of the school environment. To assist teaching staff in the planning process, the construction manuals for the developed low-cost greening systems can be downloaded free of charge from the project homepage. An input mask allows the selection between systems for indoor, open space and facade greening. Depending on the choice, a subpage opens with all relevant information on the individual greening systems and download links to the construction manuals.

## 3. Discussion

### 3.1. Greening of Schools

Making today’s schools and other educational institutions more ecological in order to anchor the idea of sustainability in the younger generation is currently an important topic, which can be seen not at least in the large number of scientific publications. However, as with any structural change, there are numerous obstacles to rapid implementation, as Jabbour, Sarkis, de Sousa Jabbour and Govindan [4] examined using two case studies in Brazil. They found that the process of mainstreaming environmental issues usually starts with research and teaching and depends on the personal motivation of a few or individual researchers.

The same phenomenon was observed in the development of the low-cost systems presented in this publication, which would not have been possible in this quality without the commitment and partly unpaid work of the teachers involved. Likewise, the implementation of greening measures at various schools depends on the initiative of the school management or an individual teacher, as long as the respective building owner (in Austria, for example, this is the Federal Real Estate Company BIG for federal schools) does not make school greening systems to a general equipment standard.

The importance of environmental education and the integration of sustainability programs in schools is also demonstrated by Denan et al. [5] using four case studies in Malaysia, Indonesia and Thailand. The findings show that many schools have already developed an awareness of environmental education. However, they argue that support from relevant authorities, potential sponsors, nongovernmental organizations (NGOs), and communities is essential for the increased use of green technology. Similarly, the environment of schools needs to be completely reformed to include climate-smart building designs and materials, and to provide sufficient green space and space for outdoor activities. This is also a particular goal of the present research project, which would not have been possible without the support of local authorities either.

A study by Cole and Hamilton [6] examines a school building before and after it is converted into a “teaching green building” with the goal of improving environmental education. A survey of green building knowledge and environmentally conscious behaviors of students in this green middle school and a reference school showed significantly higher levels of green building knowledge among students in the green school. No differences were found between the two schools in terms of environmentally conscious behavior. Here, the general school practices were of greater importance than the green building itself.

A large-scale questionnaire survey by Yamanoi et al. [7], involving over 600 elementary and secondary teachers, provided information on the factors that determine the implementation of nature-based education by science teachers. The main influencing factor identified in the survey was the degree of closeness to nature of the teachers themselves. This was followed by factors such as teachers’ ecological knowledge, frequency of nature experiences in childhood, and the environmental friendliness of the school. In order to promote nature education in schools, it is therefore important to increase teachers’ closeness to nature and ecological knowledge and to provide more green spaces in schools.

### 3.2. Plants in the School Interior

The installation of greening systems such as green walls or vertical gardens in school interiors can improve learning and increase student attention. For example, a curriculum presented by McCullough, Martin, and Sajady [8] includes the implementation of such living wall systems in classrooms to interactively connect students with nature indoors. This could provide a hands-on connection to the subject areas of science, technology, art, and mathematics, which also applies to the results of this research.

Pacini, Edelmann, Großschedl, and Schlüter [9] also show the advantages of integrating green walls into school lessons and have designed a prototypical teaching unit for this purpose that includes three different phases: a descriptive, an investigative, and a communicative phase. In the descriptive phase, an inventory of the existing green wall is made. The investigative phase follows an exploratory approach, whereby public opinion is first surveyed and based on this, scientific investigations are carried out on the green wall. In the communicative phase, the results of the previous phases are prepared in such a way that they can be presented to a larger audience. In contrast to the procedure in Section 4.1, the focus here is on the scientifically oriented investigation of greening systems, while in the present study the creative and design aspect of developing new greening solutions represents the main teaching content. Ideally, both methods are combined, so that greening systems are first developed with a school class and scientific investigations are carried out on these systems at a later stage. In this way, the topic of greening buildings could be dealt with as a class project over the course of an entire school year and thus an essential sustainability topic could be brought closer to the students in a practical way.

In addition to the added educational value of integrating indoor greening into the school curriculum, plants can also improve the indoor environment. In a case study by Danielski, Svensson, Weimer, Lorentzen, and Warne [10], a 10% lower CO_2_ concentration and slightly higher and more stable temperatures were measured in two classrooms in a secondary school in Sweden during the winter period.

### 3.3. Plants in the School Open Space

In addition to the many known positive effects of green spaces and plants on the environment, the influences on people are of equal importance. For example, Jansson, Gunnarsson, Mårtensson, and Andersson [11] studied the positive impact of greening school open spaces, especially on children up to the age of eleven. They found the greatest impact can be achieved when children were involved from early planning through ongoing maintenance of the greening. This could foster positive attitudes among children and caring behavior toward plants. At two Swedish schools, in a four-year process, children were involved in the planning, planting, management, and maintenance of the greening of the school grounds and were regularly surveyed about it. It was investigated that children’s involvement in the planning phase was crucial for the functionality of the school grounds, while children’s attitudes towards greening were determined by their long-term involvement in its management and maintenance [12]. Thus, in the present study students were involved from the planning phase to the construction of the greening systems, and they have also been introduced in the respective maintenance.

The influence of school garden on cognitive and social skills of primary school children was noted in a study by Amiri, Geravandi, and Rostami (2021). The results showed a significant difference between the students’ abilities before and after the study period. Changes in students’ attitudes towards plants, care for nature, sense of belonging to nature, and in the mood and their morals could be detected. Overall, it is postulated that school gardens are an effective educational tool that can help students develop their social behavior.

In a case study of two elementary schools in Melbourne, Australia, Onori, Lavau, and Fletcher [13] examined the various factors affecting the implementation of greening measures in schools. Four key areas have emerged for implementation:Professional roles and relationships: This includes leadership by a competent project leader, the appointment of a dedicated project manager, and the hiring of contractors with appropriate expertise (e.g., for green maintenance, irrigation system maintenance, etc.). Attention should also be paid to contracting as directly as possible and to good working relationships (including through shared sustainability values).Planning and design: criteria such as site-specific planning and design and an investment in high-quality equipment and long-term maintenance should be considered, as well as the school’s technical expertise and a timely transfer of responsibilities and expertise related to the greening effort when staff changes.Value to the school community: the focus is on disseminating knowledge about the positive effects of the implemented greening measures on the well-being and performance of the students. Further, a connection to the school’s core concerns should be done, and learning opportunities throughout all phases of the project should be identified.Engagement of the broader community: there are multiple opportunities for external support for greening projects, such as funding, resources, and expertise. This can create demonstration sites for sustainable practices and stimulate future projects in this direction. Additionally, recreational and learning opportunities can be provided for other community groups, creating a sense of community ownership for greening.

The criteria listed can also be applied to greening efforts in school interiors. Applied to the present study, the success of the process of developing and implementing low-cost greening systems can be explained by the fact that all four areas mentioned by Onori et al. were considered and optimized from the beginning (see Table 1). The result is functioning greening systems that are accepted and maintained by students and teachers and presented to the public on special occasions to motivate even more schools to develop their own ambitions for integrating greening solutions into the classroom.

## 4. Methods

### 4.1. Involvement of Students in Conceptual Design

The development of greening solutions with the direct and intensive involvement of students of the CSBT and the HBLFA, is crucial for the success of the project. Essential in this context is the cooperation with the above-mentioned schools as the primary target group. The students were able to plan the greening systems themselves, implement the first prototypes and manage them since then. At the same time, the participants were introduced to interdisciplinary work and actively participated in the design of their environment. The names of the participating students are listed on the designs and in the construction manuals, which are available for download on the project homepage.

The entire process of developing greening solutions was run through twice. In the first step, greening systems for indoor and facade application were addressed, and in the second step, systems for the open spaces of schools were discussed. In both occasions, a school class of the CSBT (a fourth and a second year class, respectively) and the HBLFA (a fifth and a fourth year class, respectively) took part and the development process was as follows (see Figure 7): Once the school classes were selected, inspirational documents for the students were created, to introduce them to the topic and presented in an online meeting by representatives of the project team. After the presentation, the students were divided into mixed groups, so that both CSBT and HBLFA were represented in each group. Each of the 12 groups was given an individual task to develop a design for a low-cost greening system for the school interior or open space (see Table 2). After a preparation period of about two weeks for research and initial idea collection, online meetings of the individual groups took place. The ideas of the groups were discussed and refined for further work. With a subsequent two-week elaboration period, the designs could be detailed in terms of construction, fastening, planting, substrate, and irrigation. The interim results were presented to the entire project team by the respective group members in an online workshop. After incorporating feedback from the project team, the first preliminary construction manuals were drafted. These served as the basis for the exemplary construction of four indoor and two outdoor greening systems at the CSBT construction yard. An overview of the final student designs is provided in Section 2.

Table 2 shows the different tasks for the three thematic groups facade, interior and open space greening systems. These were described in a PDF document and sent to the student groups. The following aspects had to be considered:sufficient size (impact on climate and/or energy use);if necessary, artificial lighting (indoors);the appropriate fertilizer;suitable substrate (soil, perlite, etc. or hydroponics);water and/or power supply;consideration of specifications regarding fire protection;budget per greening max. EUR 2.500.

In total, four tasks on indoor greening systems could be elaborated. More freedom was given in the design of the various tasks of open-space greening systems. There were two open-topic groups, whereby the above-mentioned aspects also had to be taken into account.

### 4.2. Greening Workshops at Schools throughout Austria

One of the goals of the MehrGrüneSchulen research project is the structural implementation of a low-cost greening system at one school per federal state of Austria. In order to find interested schools and to evaluate the general interest of schools in greening measures, a nationwide call for applications was sent out. Included in the schools’ application was the opportunity to submit a specific greening request. Out of 43 responses received by the deadline, nine schools, listed in Table 3, were selected that best met the requirements of the research project. Since several systems had already been planned and implemented at the CSBT, as described in Section 4.1, no other school was chosen from Vienna. For information on the greening measures implemented, see Section 2.

The greening workshops in the federal states took place between September 2021 and June 2022. Apart from the first implementation in Vienna at the CSBT construction yard, all implementations in the federal states followed the sequence shown in Figure 8: Once the participating schools were selected, initial informational materials on the developed greening systems, shown in Section 2.1 and Section 2.2, were distributed and schools were asked to indicate their preferences. After individual consultation with the schools and adaptation of the greening systems to their respective ideas and possibilities, the workshop dates were arranged and the necessary preparations, such as the completion of detailed drawings and the acquisition of the required materials and plants, were made. A basic equipment of machines and tools were prepared and provided by the responsible garden and landscape planner of the project team and taken directly to the individual workshops. The workshops themselves, which usually took one or two days, always started with an introduction to the theoretical basics and general advantages of greening systems, which was adjusted to the age and skills of the participating students at each school. Furthermore, the greening system to be erected was described and its possible effects on the school space use and design were discussed. In the next step, the supporting wooden structure for the greening systems was erected together with the students on site, under the guidance and support of craftsmen from the project team. Students were supported at every step of the process in terms of accident prevention, handling of the tools used and structural wood protection. In the same way, substrate and plants were inserted and, if necessary, climbing cords were added.

In the workshops, the participation of students did depend on their age and basic enthusiasm for handicraft work. In the implementation, the focus was on technical support and the transfer of know-how relevant to the work. At the end of each workshop, the project team handed over responsibility for the systems to the schools and provided tips and tricks for the maintenance and care of the plants and the whole system.

## 5. Conclusions

In the course of this study, a total of twelve low-cost greening systems for school interiors and open spaces were developed in an interdisciplinary collaboration between a technical school and a horticultural school, six of which were implemented and greened at the technical school’s construction yard. In addition, the constructional implementation of the adapted student designs took place at nine more schools throughout Austria, whereby school classes were involved in the construction and greening process on site.

For most of the developed greening systems, construction manuals were developed for independent replication by other interested schools and made freely available to the public. This way, the implementation of green infrastructure in Austria’s schools is being advanced and, at the same time, climate-friendly construction, the development of social and moral skills, and the relationship and interaction with nature are being promoted. This contributes to the gradual improvement of air quality and microclimatic conditions in and around schools and can increase the acceptance and awareness of building greening. In a participatory process, students and teaching staff design areas of their school premises themselves and acquire them through regular use and care of the plants, which can be seen as a continuous process of learning and thus symbolic for the school.

## Figures and Tables

**Figure 1 plants-12-00327-f001:**
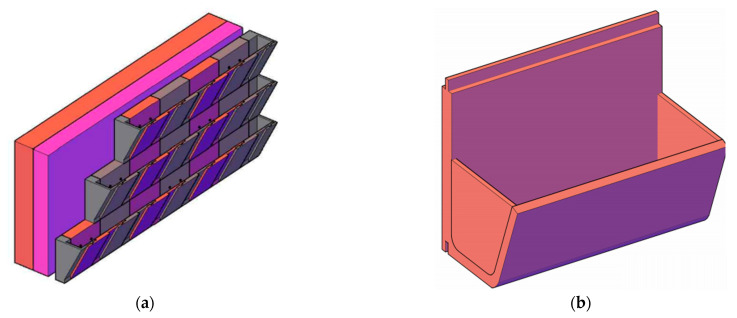
First drafts of the facade and interior greening systems of the participating students of the Camillo Sitte Bautechnikum in Vienna: (**a**) facade greening system “The Green Plug” made of clinker bricks; (**b**) facade trough made of clinker bricks; (**c**) modular wall greening system; (**d**) plantable seating furniture “Green Domino”; (**e**) plantable mobile partition wall; (**f**) greened hanging system “Green Cloud”.

**Figure 2 plants-12-00327-f002:**
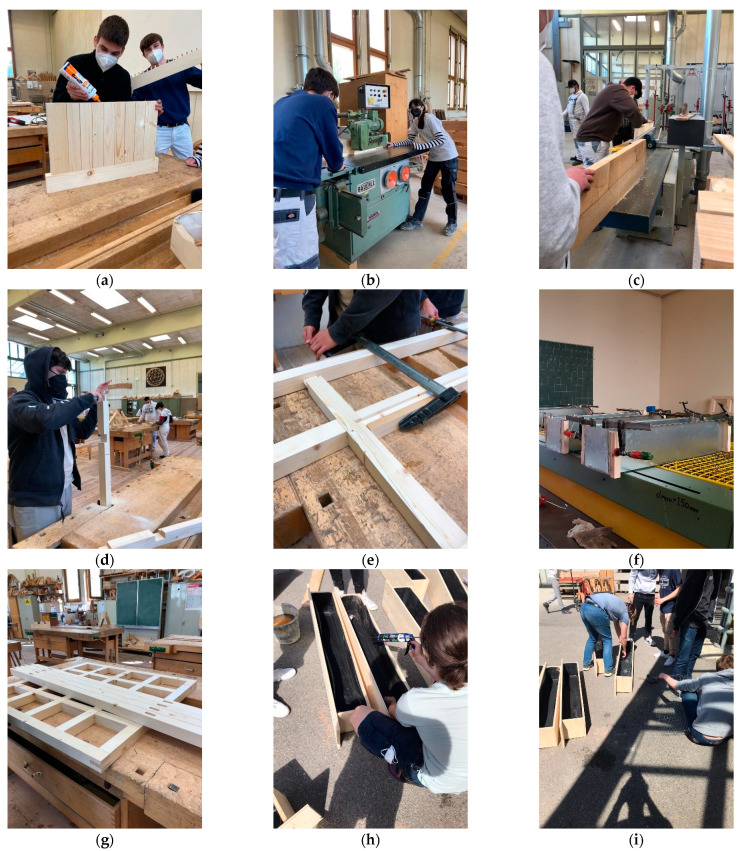
Construction of the prototypes of the indoor greening systems in May 2021 at the construction yard of the Camillo Sitte Bautechnikum in Vienna: (**a**–**c**) plantable seating furniture “Green Domino”; (**d**–**f**) plantable hanging system “Green Cloud”; (**g**) modular wall greening system; (**h**,**i**) plantable mobile partition.

**Figure 3 plants-12-00327-f003:**
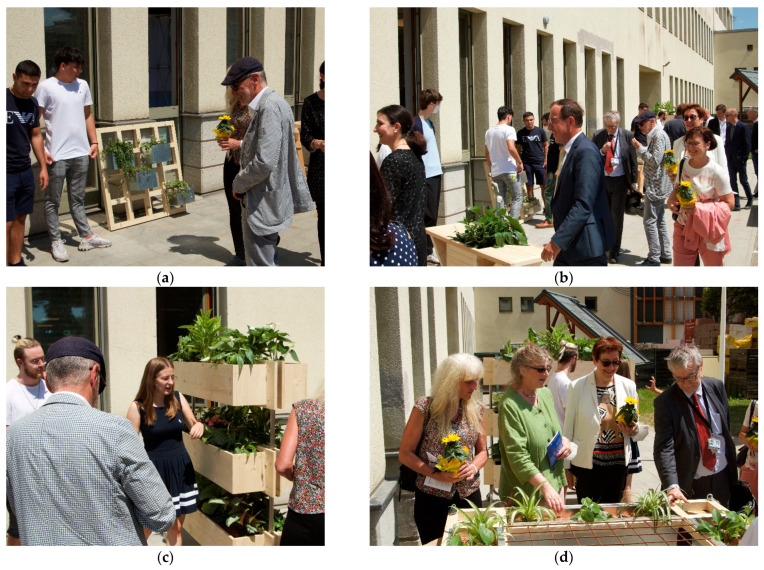
Presentation of the prototypes of the indoor greening systems at a public event in June 2021 at the Camillo Sitte Bautechnikum in Vienna: (**a**) modular wall greening system; (**b**) plantable seating furniture “Green Domino”; (**c**) plantable mobile partition wall; (**d**) plantable hanging system “Green Cloud”.

**Figure 4 plants-12-00327-f004:**
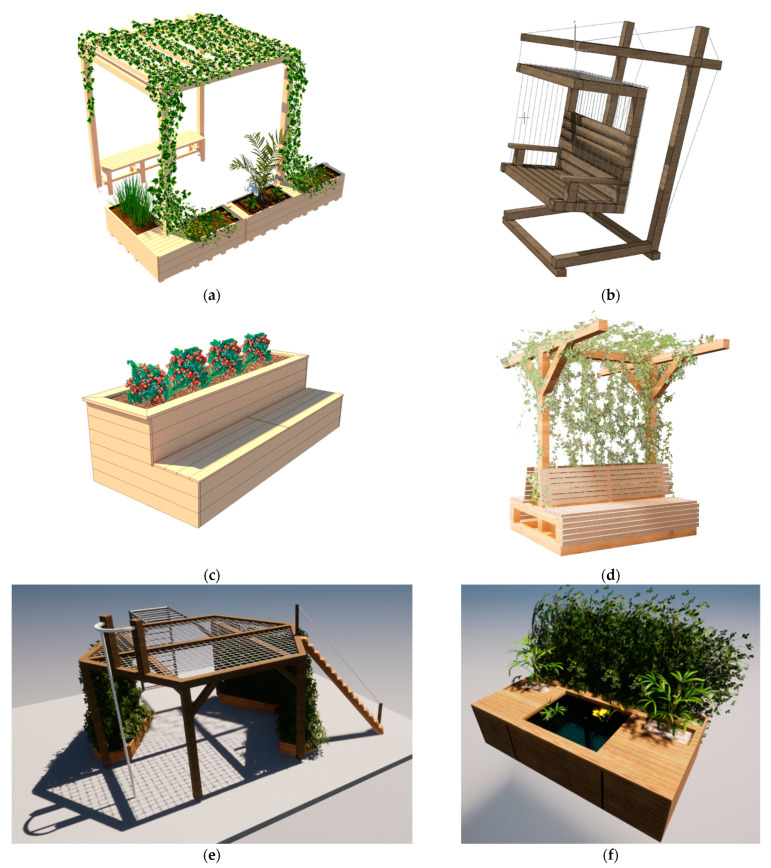
First drafts of the open space greening systems of the participating students of the Camillo Sitte Bautechnikum in Vienna: (**a**) greened pergola “GreenClassroom”; (**b**) plantable hanging chair “Green Trio”; (**c**) raised bed “The Vessel”; (**d**) pergola “T-Bench”; (**e**) pergola with playground equipment “Place Evergreen”; (**f**) greened fountain.

**Figure 5 plants-12-00327-f005:**
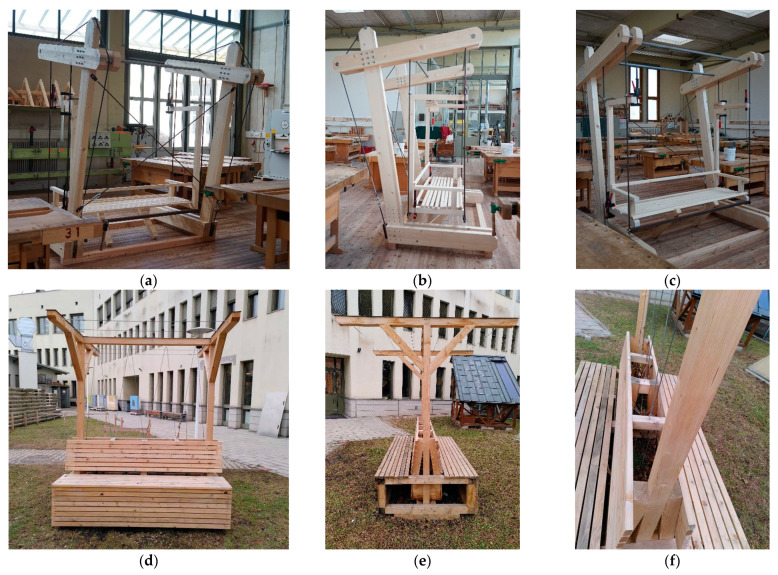
Open-space greening systems built by students at CSBT in Vienna: (**a**–**c**) hanging chair “Green Trio”; (**d**–**f**) pergola “T-Bench”.

**Figure 6 plants-12-00327-f006:**
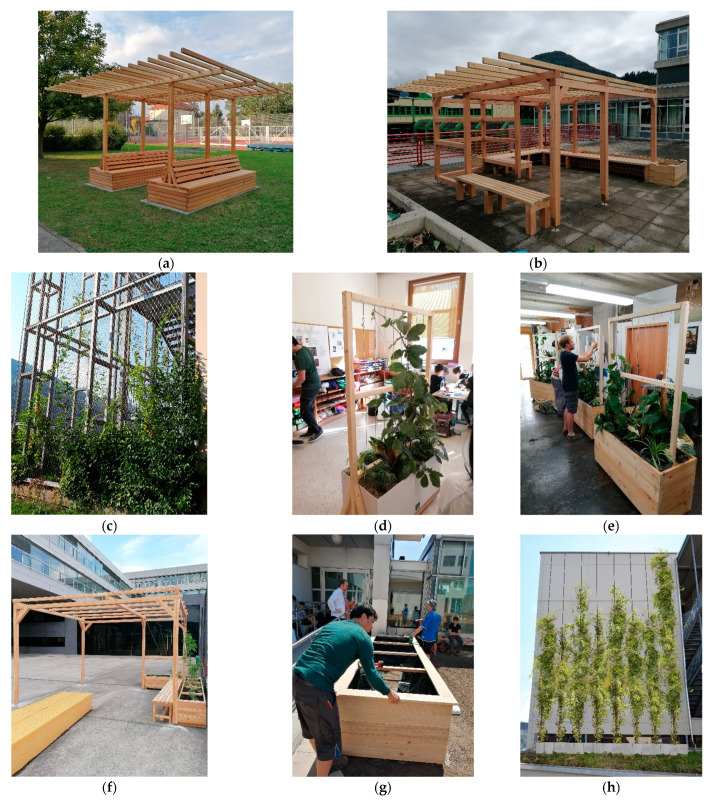
Results of the greening workshops in Austria’s provinces: (**a**) Korneuburg (Lower Austria)—pergola “T-Bench”; (**b**) Wörgl (Tyrol)—pergola “Green Classroom”; (**c**) St. Johann (Salzburg)—climbing plants; (**d**) Maria Gail (Carinthia)—mobile green wall; (**e**) Graz (Styria)—mobile green wall; (**f**) Kirchdorf (Upper Austria)—pergola “Green Classroom”; (**g**) Neusiedl (Burgenland)—raised bed; (**h**) Lauterach (Vorarlberg)—climbing plants.

**Figure 7 plants-12-00327-f007:**
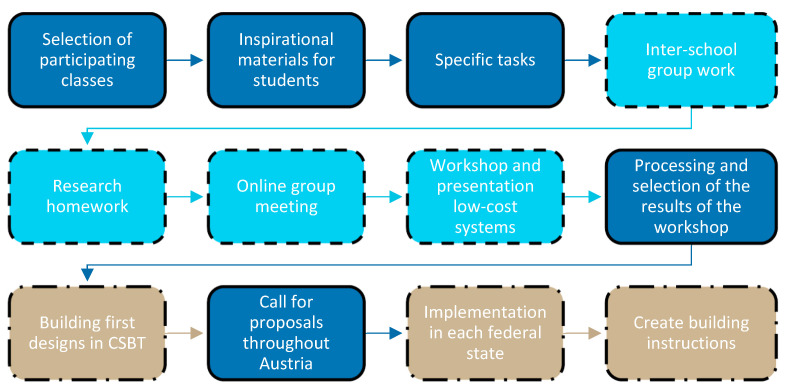
Development process of low-cost greening systems; dark blue, framed throughout: work in the project team; light blue, dashed framed: integration of the school classes of the CSBT and the HBLFA; beige, dash dot framed: joint work of the project team with students of the CSBT and other selected school classes throughout Austria.

**Figure 8 plants-12-00327-f008:**
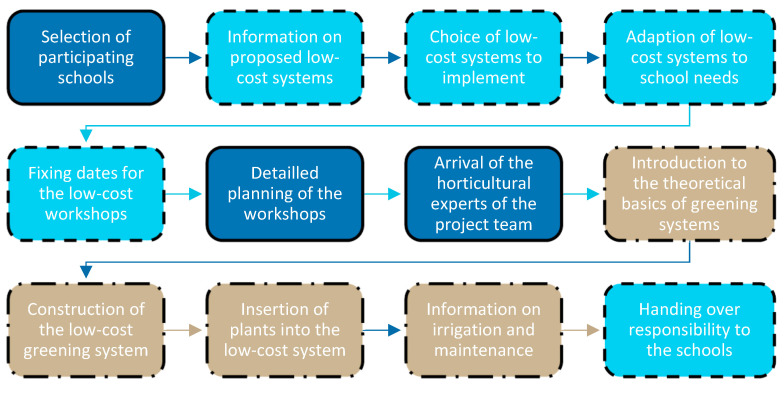
Realization of the low-cost workshops in the federal states; dark blue, framed throughout: work in the project team; light blue, dashed framed: integration of the school authorities; beige, dash dot framed: integration of the students of the selected school classes throughout Austria.

**Table 1 plants-12-00327-t001:** The four influencing factors for the implementation of greening measures according to Onori et al. [13] applied to the implementations at Camillo Sitte Bautechnikum.

Factor	Implementations at CSBT
Professional roles and relationships	The project management was carried out by the Vienna University of Technology—Research Department for Ecological Construction Technologies, as the necessary know-how and resources were available. The expertise for the practical implementation of the greening systems was available at the CSBT’s construction yard. The definition of the planting, the irrigation, and the substrate was done by the HBLFA. The construction workshops in the provinces were led by the garden and landscape planner Dipl.-Ing. Ralf Dopheide e.U.
Planning and design	The planning and design were done by the participating students of the CSBT and the HBLFA. The planning was directly oriented to the needs of schools. A school or teaching staff with an affinity for greenery is responsible for maintenance.
Value for the school community	Four different school classes and numerous teachers from the participating schools and the building yard at the CSBT were involved in the development and subsequent constructional implementation of the greening systems. The finished systems are set up in the auditorium of the CSBT and are thus also available to future classes as teaching material or as a basis for student’s theses.
Engagement of the broader community	The development of the greening systems was carried out within the framework of the research project “MehrGrüneSchulen” and was supported by this project in an advisory manner. The completed indoor greening systems were presented to the relevant interest groups and stakeholders in the school sector as well as to the funding agencies at a public school event. Since then, more visits from other interest groups have taken place.

**Table 2 plants-12-00327-t002:** Assignments for participating school classes to design low-cost greening systems for school interiors or open spaces.

Nr.	System for	Task/Topic
1	Facade	Continuation of the “facade stone” project for north-facing facade
2	Facade	Continuation of the “facade stone” project for south-facing facade
3	Interior	Modular system made of recycled materials
4	Interior	Suspended modular system—for aisle areas
5	Interior	System with a seating option—for auditorium or dining room
6	Interior	Mobile system as green partition wall
7	Interior	Ceiling suspended system—for a south-facing class
8	Open space	Green pergola
9	Open space	Climbing plants
10	Open space	Raised bed
11	Open space	Shading through greenery
12	Open space	Open topic
13	Open space	Open topic

**Table 3 plants-12-00327-t003:** Schools selected for implementation of low-cost greening measures.

Nr.	Federal State	City	School
1	Vienna	Vienna	Technical college (CSBT)
2	Burgenland	Neusiedl	Business and tourism school
3	Carinthia	Maria Gail	Elementary school
4	Lower Austria	Korneuburg	Commercial college
5	Upper Austria	Kirchdorf	High school
6	Salzburg	St. Johann	Commercial college
7	Styria	Graz	High school
8	Tyrol	Wörgl	Commercial college
9	Vorarlberg	Lauterach	High school

## Data Availability

The data presented in this study are openly available at the TU-Wien project homepage at https://www.obt.tuwien.ac.at/mehrgrueneschulen/begruenungssysteme-und-finanzierung/ (accessed on 28 November 2022).

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
