# Peer review of "Cross-School Collaboration to Develop and Implement Self-Construction Greening Systems for Schools"

_plants, 2023, doi:10.3390/plants12020327_

Round 1
Reviewer 1 Report
Good quality article, the topic is relevant not only for a broad academic community, but also for practitioners and other relevant stakeholders. The structure of the article is consistent with the requirements (the abstract disclose the contents of the main research results; the illustrations (tables, figures) are relevant). The article includes enough scientific analysis of literature and other sources on the subject of research. The research methods used are dequate and sufficient for this topic the results of the research reliable and well-founded. The conclusions in the article correct and logically grounded.
Author Response
Thank you for reviewing our manuscript. We did some improvements according to the other reviewer’s comments and are looking forward to a soon publication.

Reviewer 2 Report
Although the article presents the results of a participatory approach for greening the school space to mitigate climate change impacts by harvesting the carbon emissions in the space of the school domain, which is fine and worth considering its review for possible publication, the methodology is not presented. Therefore, the corresponding author needs to revise the article by including the (2. Methodology or 2. Materials and Methods) in detail in the article. Also, a detailed flowchart for the methodology should be added in the methodology section.
Author Response
Thank you for reviewing our manuscript. We did improve the methods in section 4 and did include another flow chart in section 4.2 according to your comments. Due to the fixed order of the individual sections given by the journal template, we could not change the methods section to section 2. We kindly ask for your understanding.

Reviewer 3 Report
1. ArtykuÅ‚ ma charakter prezentacyjny. W treÅ›ci publikacji przedstawiono metody pracy z dziećmi w wieku szkolnym oraz wyniki pracy warsztatowej uczniów i ich praktyczne zastosowanie. Sami autorzy stwierdzajÄ…, że "celem artykuÅ‚u jest pokazanie procesu tworzenia systemów zazieleniania w ramach interdyscyplinarnej formy nauczania oraz przedstawienie wyników pracy studentów i warsztatów budowlanych oraz ekologicznych".
2. Warsztat badawczy nie zostaÅ‚ moim zdaniem odpowiednio przedstawiony. JeÅ›li celem badaÅ„ miaÅ‚o być pokazanie procesu tworzenia systemów zazieleniania w ramach programu badawczego "MehrGrüneSchulen", na tle doÅ›wiadczeÅ„ innych badaczy, które autorzy cytujÄ… w rozdziale "3. Dyskusja", to rozdziaÅ‚ ten należaÅ‚oby uzupeÅ‚nić o porównanie prezentowanych wÅ‚asnych metod badawczych i ich wyników z wynikami opisanymi przez wspomnianych badaczy.
3. Układ treści publikacji jest nieprawidłowy. Powinien zawierać następujące rozdziały w przedstawionej kolejności:
1. Wprowadzenie
2. Cel badań
3. Przedmiot badań
4. Metody
5. Wyniki
6. Dyskusja
7. Wnioski
Author Response
Thank you for reviewing our manuscript. We did improve the description of the research workshops in section 4. Methods according to your comments. Also, the comparison of the presented own research methods and their results with the results described by the mentioned researchers was complemented in section 3. Discussion.
Due to the fixed order of the individual sections given by the journal template, we could not change the layout of the content of the publication according to your recommendations. We kindly ask for your understanding.

Round 2
Reviewer 2 Report
I have not further comments. It is largely improved.